# Technology-Based Substance Use Interventions: Opportunities for Gender-Transformative Health Promotion

**DOI:** 10.3390/ijerph17030992

**Published:** 2020-02-05

**Authors:** Julie Stinson, Lindsay Wolfson, Nancy Poole

**Affiliations:** 1Centre of Excellence for Women’s Health, D404-4500 Oak St, Vancouver, BC V6H 3N1, Canada; lindsay.wolfson@gmail.com (L.W.); npoole@cw.bc.ca (N.P.); 2Canada FASD Research Network, PO Box 11364, Vancouver, BC V5R 0A4, Canada

**Keywords:** gender, technology, gender transformative, health promotion, substance use, SGBA+, substance use prevention

## Abstract

Drawing on data from a scoping review on sex, gender and substance use, this narrative review explores the use of gender-informed and technology-based approaches in substance use prevention and health promotion interventions. With an ever-changing landscape of new technological developments, an understanding of how technology-based interventions can address sex, gender, and intersecting equity considerations related to substance use is warranted. Current technology-based approaches to substance use prevention and health promotion are described and assessed for gender-specific and gender transformative outcomes, and limitations are discussed related to inclusivity, access, confidentiality, and a dearth of research on technological approaches that integrate gender-based analysis. A call for action designed to advance technology-based health promotion, prevention and brief interventions that address gender equity simultaneously with substance use is proposed.

## 1. Introduction

Historically, substance use prevention and treatment interventions have employed directive, gender-blind and abstinence-oriented approaches [1,2,3,4,5,6,7]. However, research from the substance use field has demonstrated the importance of sex and gender considerations in substance use responses including prevention and treatment interventions [8,9,10,11]. The gendered factors, influences, and differences as to how individuals respond to substance use prevention and cessation cannot be ignored. For example, concerns surrounding weight gain have been found more often to be a reason for avoiding smoking cessation for women compared to men [12]. Other gender-mediated reasons for smoking cessation have demonstrated that men are more likely to quit as a result of tobacco policy, and that social unacceptability is more closely associated with quitting among women [13]. Such gender-informed influences are integral to understanding how to approach substance use health promotion, prevention and brief intervention efforts.

In understanding gender-responsive approaches, it is important to consider the range in which gender inclusions and considerations can impact gender equity outcomes. See Figure 1. Gender-blind programs ignore gender norms, roles and relations and may therefore reinforce gender-based discrimination, biases and stereotypes. Gender-specific programs acknowledge gender norms, consider women’s and men’s specific needs and act to accommodate these needs to some degree. Gender-transformative approaches focus on the dual goals of improving health, social or economic status as well as gender equity [14]. The benefits of gender-transformative approaches have been demonstrated globally, addressing gender norms, stereotypes or relations as a route to improved health outcomes when undertaking prevention and health promotion efforts [14,15,16,17]. Future programs and studies can be created or evaluated using sex and gender-based analysis plus (SGBA+) to determine their implementation (or lack of) of a gender-transformative approach or to analyze the effectiveness of programs for promoting inclusivity. This form of analysis provides an important perspective to evaluating programs and evidence for potential gaps in relation to sex, gender and intersectional equity [18].

Simultaneous to the growing commitment to SGBA+, there has been an increased interest in incorporating technology into substance use prevention and treatment. Technology has become more interwoven into day-to-day life and into the health care sector with the advancements of technology and Internet-based care (eHealth). In one regard, technology provides a significant opportunity for prevention and health promotion interventions to reach large audiences. Technology has more recently been used to connect with adolescents and young adults but also older generations and individuals of all genders, with a focus on health promoting and harm reducing decision making. However, gaps and cautions remain as to the reach and inclusivity of these technological approaches to date. Given the dearth of research beyond web-based tobacco and alcohol interventions [19], an understanding of the scope of technological approaches to prevention and health promotion for all substances is warranted. Further, it is timely to examine how sex and gender play into decision making surrounding substance use prevention efforts and how gender-informed harm reduction-oriented approaches might be further or more effectively achieved in technology-based interventions. This article will provide an examination of how technology-based substance use prevention approaches are integrating health promotion and harm reduction, and how sex, gender and intersecting factors of age, race, and socioeconomic status related to substance use are, or are not, being achieved in these efforts. It will further provide recommendations for the direction of future research on technology-based substance use prevention and health promotion approaches.

## 2. Methods

This narrative review is based on a subset of findings from a scoping review conducted on the integration of sex and gender in research on opioids, alcohol, tobacco and cannabis prevalence and patterns of use, health effects, and prevention, treatment and harm reduction interventions and outcomes. The four substances were chosen for having differential sex and gender impacts, and for their current critical concern in Canada.

The scoping review was based on two broad questions:(1)How do sex- and gender-related factors impact: (a) patterns of use; (b) health effects of; and (c) prevention/treatment or harm reduction outcomes for opioid, alcohol, tobacco/nicotine and cannabis use?(2)What harm reduction, health promotion/prevention and treatment interventions and programs are available that include sex, gender and gender-transformative elements and how effective are these in addressing opioid, alcohol, tobacco/ nicotine and cannabis use?

The scoping review methodology has been described in full in Hemsing et al. [20]. Over five thousand (5030) papers were included in the original review, which excluded research specific to pregnant women who use substances, but was inclusive of women, girls, men, boys, trans and gender diverse people of all other ages and demographics. The included papers were categorized by substance (alcohol, tobacco, opioid, and cannabis and multiple substances/substance use generally) as well as by research focus (i.e., prevalence, health effects and biological responses, and prevention and intervention type) in an Endnote library. This narrative review draws upon articles from the scoping review findings. The authors reviewed articles categorized in the prevention (64 articles), harm reduction (50 articles), and brief intervention (76 articles) fields for relevance and did additional key word searches in the full library using the terms: Tech(nology) (8 articles), Health Promotion (8 articles), Web(site) (15 articles), Internet (8 articles), App(s) (3 articles), Computer(s) (11 articles), Mobile (4 articles), Text Message (2 articles), (Tele)Phone (9 articles), Online (11 articles), Social Media (5 articles), Digital (3 articles), Fitbit (1) and Video(s) (3 articles). Due to the prevalence of articles that focused on technology-based brief interventions, an additional targeted search was done for the studies categorized as having a research focus on brief interventions, which identified 17 papers. 52 articles from the combined search were excluded for being duplicate values.

A total of 239 articles were analyzed for inclusion, 54 meet the criteria of examining sex and/or gender in technology-based substance use prevention approaches and outcomes, and 45 were included in this article. Of these articles, 11 were computer-based, 12 were phone-based, 19 were web-based and 3 examined multiple technological mediums. Since ‘technology-based interventions’ was not a specific category in the original scoping review, a narrative review approach was used for this article in order to examine all of the literature more broadly related to this topic.

## 3. Technology-Based Substance Use Prevention and Health Promotion Approaches

There were multiple ways in which substance use prevention and health promotion approaches incorporated gender and technology in the literature including brief web-based interventions (16 articles), targeted web-based messaging (three articles), social media, websites and virtual communities (eight articles), computer games (three articles), text messaging (three articles), mobile phone applications (apps) (seven articles) and telephone interventions (five articles). Of the 45 included articles, 11 reported on patterns and prevalence of use and 34 evaluated the efficacy of technology-based interventions. The interventions ranged from gender blind to gender transformative. Of the nine articles that described gender-transformative approaches, only five unique interventions were reported—all of which were conducted with only girls or women. Two of these interventions were brief web-based interventions, two were mobile phone apps, and one was a telephone-delivered intervention.

As a result of the dearth of gender-transformative interventions exemplified in the literature, the review findings have been categorized by technological approach in order to examine the literature more broadly and provide a critical analysis of how gender has been considered across all technology-based platforms (see Appendix A for details). The following section will examine gender inclusion by technological medium and how these platforms have incorporated variances of harm reduction, health promotion, prevention, and treatment.

## 4. Gender Considerations in Technology-Based Substance Use Interventions

### 4.1. Brief Web-Based Interventions

Brief web-based interventions represented web/computer-based programs often in module format that embodied an education and/or health promotion approach to substance use harm reduction. They were the most advanced in taking a gender-informed approach, though studies were often only conducted with single gender groups of girls or women. For example, one gender-transformative web-based brief intervention focused on girls (12–13 years old) and the relationship between mothers and daughters as a mechanism for health promotion and substance use prevention [21], promoted social skills (self-efficacy, communication) and decision making as ways of coping with stress, managing substance use, questioning peer norms, and understanding media influences [22]. This home-based module program was conducted on the internet or with a CD-ROM on gender-specific substance use prevention topics guided by family interaction theory, completed by mother–daughter dyads. Dyads were advised to complete one 45 min session/week [23]. Results showed stronger communication and closeness between the mother and daughter, stronger substance use refusal skills for girls, and improved parental monitoring and rule setting in regards to substance use [21,22,24]. Girls also showed improved conflict management, improved problem-solving skills, and reduced stress [21,24]. One of the most significant differences was the change in normative beliefs among the girls, and reduced substance use overall [24]. The results remained constant at one- and two-year follow ups, with decreased substance use and intention to use among girls who were involved in the program, as well as lower weekly alcohol consumption among mothers at two-year follow-up [24,25].

This program was tested with a variety of populations of mothers and daughters. When tested with girls aged 10 to 12 and their mothers who reside in public housing, similar positive results were seen, with mother–daughter communication and closeness, reduced stress, and stronger refusal skills reported by girls at 5 month follow up [21]. Similarly, with a population of Hispanic and African American girls aged 10 to 13 and their mothers, the intervention resulted in overall increased protective factors and reduced risk factors in connection to communication, knowledge and refusal skills. It also resulted in lowered girls’ alcohol use and intention to use in the future for all substances [23]. When conducted with a population of Asian-American adolescent girls, results showed reduced depressed mood and reduced substance use/intention to use in the future [26]. Overall, this program, based in attachment and nurturing parental relationships as a means of promoting healthy behaviour, showed both change in behaviour and beliefs, as opposed to change in beliefs only. This web-based model also enables girls to engage with the program at their own pace and in their location of choice [22]. A similar gender-specific intervention, called RealTeen, that involved 13- and 14-year-old girls, rather than the mother–daughter dyad, also reported lower rates of substance use among girls, healthier normative beliefs, and stronger self-efficacy as they relate to substance use [27].

Other brief web-based interventions designed to reach and engage women often combined interventions on substance use with other gender-based health concerns such as sexual or relational violence [28,29]. For example, Gilmore et al. examined the impact of a combined brief web-based substance use intervention and sexual assault risk reduction intervention for college women [29,30]. The authors found that offering a dual intervention provided college-aged women with a comprehensive set of skills, resources and protective behavioural strategies to both reduce heavy episodic drinking and risk of sexual violence [29,30]. Another gender-transformative program, ‘BSAFER’ (Brief Intervention for Substance Use and Partner Abuse for Females in the Emergency Room), was designed for women who had accessed the emergency department for substance misuse and intimate partner violence. This brief web-based intervention was conducted on a tablet during an emergency department visit and included goal-setting questions and informational videos. The intervention resulted in decreased weekly substance use [28]. However, despite favourable results in reduced heavy drinking, the authors noted that the effectiveness of the program may be circumstantial to women with higher risk severity for sexual assault and intimate partner violence [28].

There were gender-sensitive interventions that reported gender differences in the uptake of brief web-based interventions. For example, in one study Hispanic men tended to respond more favourably to web-based brief interventions, compared to Hispanic women, who preferred in-person brief interventions [31]. Another study, focused on drinking self-assessment, found that the intervention appealed more to women, despite no significant differences in drinking outcomes between men and women (the intervention resulted in reduced consumption and reduced consequences from drinking for men and women) [32].

Many studies did not report on tailored gender-specific approaches, rather they reported on the gender differences in responsiveness to brief web-based interventions that were gender blind in design. However, the brief interventions described show positive results in reducing substance use and related health concerns for women and girls, when they intentionally address key gendered influences on substance use or protective factors known to be specifically helpful for girls/women.

Brief web-based interventions with personalized normative feedback have been able to address misperceptions about substance use norms by bringing in self-reflection, connecting an individual’s harmful substance use to ‘atypical’ as opposed to ‘normative’ beliefs [33,34]. This method has resulted in reduced alcohol consumption, and changes in perceived norms, for both men and women, though with varying results when integrating gender-specific feedback [33,35]. Lewis et al. found that gender-specific normative feedback was effective for women who identified more strongly with their gender [35]. This finding demonstrates that perceived gender norms are a mediating factor for how much women drink, and that gender-specific programming works for women. These programs provide evidence that gender identity is a determining factor of the success of brief web-based interventions for substance use, which further evidences the need for gender-transformative programming. However, more work is warranted to demonstrate the efficacy of web-based normative feedback for men and gender-diverse individuals.

### 4.2. Virtual Communities, Social Media and Targeted Web-Based Messaging

Other computer-based interventions in the literature included the use of websites, virtual communities, and social media in substance use interventions. Websites dedicated to substance use prevention—most commonly smoking cessation—and/or health promotion were not as commonly researched or evaluated. Those included in the review appeared to be out of date for the target demographic’s preferred technological delivery modality or they reinforced gender-stereotypes as opposed to being gender transformative [36]. Virtual communities and social media interventions were comparatively gender-informed [37,38].

Virtual communities primarily included chat forums as peer-support models (commonly located on prevention and harm reduction websites). Virtual communities and social networking chat fora have been correlated with increased alcohol use; a finding stronger among girls compared to boys [39] demonstrating that girls may be more susceptible to engage in harmful drinking via online pressures. This gender difference is especially present among adolescents at high risk who lack healthy coping strategies [39].

Social media approaches involved the creation of pages or purchased advertising space on social media platforms to relay prevention and health promotion messages, such as the health benefits incorporated with abstaining from smoking [38,40]. Targeted web-based messaging used similar approaches to social media campaigns but expanded beyond social media platforms to include all web-based platforms [41,42,43]. Both social media platforms and targeted web-based messaging have often been used as opportunities for marketers to promote alcohol use, which has resulted in increased substance use particularly among young people [39,44]. However, when used as a medium for prevention and health promotion messaging, these mediums have also been proven to increase smoking cessation and awareness of health effects of smoking and secondhand smoke, including its connections to breast cancer for both boys and girls [37,38,40,41,42]. Richardson et al. examined gender- and culture-specific web-based messages surrounding tobacco smoke and breast cancer. They found that Indigenous boys and girls involved the study were more likely to agree that secondhand smoke increased risks of breast cancer, compared to a control group that was presented a gender-neutral visual messages [41,42]. Participants also spent significantly more time viewing the messages and girls who received the gender- and culture-specific intervention were 52% more likely to request additional information about secondhand smoke and breast cancer, compared to those in the control group [41]. Though knowledge increased, messaging did not change smoking status, intention to try smoking, or intention to avoid second hand smoke [42].

### 4.3. Computer Game Programs

Computer game interventions used simulation to increase understanding of the effects of substance use, refusal skills, or changing patterns and behaviours [45,46]. Few computer game programs integrated gender considerations in design, delivery or specific audience to be reached. One game entitled Guardian Angel, though gender blind in design, was tested on male veterans to encourage practice of relapse prevention skills using simulation. Participants would make daily decisions to support the recovery of a simulated character, designed to address coping skills and high-risk environments. The findings demonstrated reductions in binge drinking and higher self-efficacy [45]. Another game, Game On: Know Alcohol (GO:KA), combined social marketing and education, alternating game and knowledge-based modules focused on a harm reduction approach. GO:KA demonstrated one of the more successful interventions, that effected behavioural change as opposed to an attitude change only [43]. However, while GO:KA did show attitude changes for boys and girls, behaviour changes and reductions in alcohol use/intended alcohol use were only evident for girls, and the intervention design was not gender informed [43]. Other studies have also shown education-based programs to be particularly effective in changing girls’ substance use behaviours [43,47,48], but not boys. These gender differences in behavioural change could be connected to evidence suggesting that education-based programs need to be repeated in more long-term settings to be effective [43,47,48].

### 4.4. Mobile Health Applications (mHealth Apps)

Mobile health apps were more commonly described in the literature as traditional ‘mHealth’ approaches, where an application on a smartphone monitors certain activity and provides alerts based on behaviours. Much like step trackers or heart rate monitoring apps, the mobile apps focused on substance use behaviours, such as alcohol frequency or drinks consumed [49]. MHealth apps provide some of the leading examples of technological innovation, including in their ability to engage individuals in substance use prevention and health promotion [49]. However, the literature has described the functionality, technical issues, and security and privacy of the apps as key barriers and concerns to programmatic success [49,50]. In regards to gender-informed approaches in intervention design, programs have ranged from gender blind to gender transformative, with some reporting gendered differences to uptake and success.

The gender-transformative program See Me Smoke-Free addresses smoking, diet and physical activity among women who smoke. Findings showed significant increases in healthy eating and physical activity, as well as higher rates of smoking cessation [51]. However, for men, similar to other technology-based programming, mobile health apps did not appear to be as successful. For example, the Swedish app *Promillekoll* created to reduce heavy episodic drinking through awareness by allowing users to test blood alcohol content levels showed men increasing drinking frequency while using this app, which was not the case for women [19]. This may have been a result of the program design being gender blind or, by comparison, the increased use and interest by women in using technology for substance use interventions [52,53].

Apps in the form of wearable tech, such as with a Fitbit, can also provide health promotion and support for transitional recovery as people leave intensive inpatient treatment. Abrantes et al. tested a gender-transformative lifestyle activity program among women with depression involved in inpatient treatment for alcohol. In-person physical activity counseling during inpatient treatment, combined with a Fitbit to monitor a step-counting goal, resulted in 44% of women abstaining from alcohol during the 3 month program. The authors suggested testing the usefulness of this health promotion intervention with women as they transition from treatment to community [54].

### 4.5. Text Message

Text message programs [55,56,57] used short message service (SMS) to send harm reduction messages related to substance use to participants. While some text-based programs were person-assisted, where an in-person brief intervention was supplemented with two-way text message goal planning, others were more directed by technology, with participants receiving pre-programmed messages on personal control and behavioural change [55,56]. The text message programs primarily focused on brief alcohol interventions for socioeconomically disadvantaged men or for men and women on college and university campuses [56,57]. In the studies with men who were socioeconomically marginalized, the men in the intervention group tended to engage at high levels with the program and appeared to be more comfortable in regards to divulging sensitive personal information about their relationships and experiences with harmful drinking with this format compared to others [55,56]. In the mixed-gender campus study, setting drinking-limit goals via SMS were found to promote a decrease in drinking particularly for men in comparison to women [57]. Though the intervention for socioeconomically disadvantaged men describes some program adaptation to intersectional context in regards to topics and language, it lacked detailed solutions to inequities and harmful norms, falling more in line with a gender-sensitive approach. The campus-based intervention could be considered gender blind, lacking acknowledgement of any gender considerations in programming. This was a common occurrence in the literature, whereby gender was not considered in the intervention design or was not explained in the study findings beyond the gender disaggregation.

### 4.6. Telephone-Delivered Interventions

Telephone-delivered interventions that were gender informed were most commonly reported for women as alcohol [58,59] and tobacco [12,60] cessation interventions. Phone-delivered interventions, including quitlines, have shown some promising results in connection to women over the age of fifty, specifically for alcohol reduction; however, telephone-delivered interventions included in the review rarely used a sample other than women, and when they did the intervention was gender blind [58,59,61]. One qualitative study examining binge drinking, depression and Post Traumatic Stress Disorder (PTSD) among women veterans found the women showed high acceptability to using phone-based care for health monitoring, feedback, and counseling [58]. This may be because of the culture and stigma surrounding substance use and gender [61]. For example, Kim et al. describe the reluctancy of women of Korean ethnicity to pursue in-person smoking cessation treatment due to the stigma attached to women who smoke, and how telephone and online programs that can be accessed privately in their homes are preferred. In this study, the younger women preferred a videoconferencing intervention over a phone-based model [61].

Phone-delivered interventions may be preferred among women of older generations due to their longer lifetime experience with this mode of technology, compared to newer forms such as mobile health apps. The longer length of the phone-delivered interventions may be too time consuming for younger women who have grown up with the internet and fast-paced technology all their lives [58]. Keeping with the pace of technology may also be a reason why phone-delivered interventions did not present as commonly used methods. However, considerations should be made to ensure that individuals who find phone-delivered interventions to be most effective can still be reached.

## 5. Future Directions for Gender Integration in Technology-Based Substance Use Interventions

### 5.1. Gender-Transformative Approaches

How gendered approaches have been integrated into substance use prevention and harm reduction efforts can be seen in the light of the gender response continuum—ranging from gender blind, to gender sensitive/specific, to gender transformative [14] whereby gender is ignored, accommodated to lesser or greater degree or transformed through attention to gender equity. Although progress has been made, particularly with girls in regards to gender-informed technology-based substance use programming [48], some programs have continued to reinforce negative gender stereotypes, such as focusing on heteronormative attractiveness as a reason for reducing consumption or positing sexual assault reduction as women’s and girls’ responsibility, as opposed to promoting health and critical thinking [27,36]. Although the effectiveness of programs in relation to the gendered outcomes achieved was explored in many of the studies, there continues to be a lack of programming that is tailored to address specific gendered influences on substance use, or that attempts to address some aspects of gender equity while also addressing substance use prevention or harm reduction (i.e., that is gender transformative). Due to its reach and low-cost in comparison to other substance use interventions, the online environment provides a key location for offering gender-transformative interventions.

### 5.2. Gender and Intersecting Equity Considerations

A lack of a gender-informed or transformative approach is paired with a lack of attention to other aspects of equity. These would be mitigated if SGBA+ had been applied in the design and delivery of programming. For example, although over 70% of young women look to the internet for health and substance use related prevention or cessation information, women of colour are less likely to seek out this information online [36,62]. Other findings suggest that, regardless of age, Caucasian men and women were more likely to use a smoking cessation website compared to men and women of colour, with the exception of a virtual community forum used by Hispanic young adult men and women [36].

The need for attention to interventions that move beyond the gender binary as well as consider sexual orientation and culture in the intervention design and implementation is also important as the prevalence of smoking among LGBTQ+ identified youth and young adults is significantly higher than among youth and young adults who identify as heterosexual [50]. The larger scoping review, from which the literature for this review was derived was inclusive of women, girls, men, boys, trans and gender diverse people of all ages and demographics, excluding pregnant women. However, there were no technology-based substance use interventions that were inclusive of or targeted to trans and gender diverse people, beyond one study that examined LGBTQ+ youth and young adults’ perceptions of a culturally tailored mHealth app for smoking cessation. In this study by Baskerville et al., LGBTQ+ identified youth and young adults have expressed that substance use mobile health apps that are culturally-tailored and recognize their specific needs are highly valued when seeking out substance use prevention programming [50]. This has also been reflected in studies that have incorporated elements of Indigenous culture into interventions with Indigenous youth, which have seen that the boys and girls prefer this type of format, are more inclined to receive the messaging, and girls in particular are more inclined to want to learn more [41,42,47].

### 5.3. Gender, Inclusivity and Access

Technology-based interventions must consider gender, inclusivity and access to technology. Often individuals have differing and inequitable access to technology; however, participants in technology assisted programming are required to have their own access to a computer, the Internet, or a phone with data. In one study, although 80% of participants said they would be interested in participating in a mobile health treatment intervention, there was differing interest by gender—with women being more interested than men—and only 50% of interested participants indeed had access to data on their cellphones [52].

Age must also be considered. Age has shown to be a factor not only with technology use in general, but for how and what it is used for. For example, one study found that younger adults (aged 18–34) visited a smoking cessation website significantly less often than adults over 34 years old, viewed fewer pages, and were less likely to use the virtual community page [36]. Consistent with other research findings, young men in particular were less likely to use the website compared to young women [36]. This could be related to how younger generations, who have grown up on faster and more immediate technology-based resources, have considerably different knowledge and methods of using technology, use it at a different pace, and use considerably different platforms. These reflections of age, gender, and how they affect use should be taken into account when designing and implementing technology-based substance use prevention and health promotion initiatives.

### 5.4. Gender, Technological Interventions and Confidentiality

There is concern surrounding the collection of data by mobile technology. Studies have shown that both men and women have expressed concerns pertaining to how their private information will be used or communicated outside of face-to-face interactions [50,53,58]. These concerns are reflected in how individuals engage with technology-based programs. For example, female veterans have expressed reluctance to share their information, viewing the phone-delivered programming as ‘too impersonal’ [58] while younger men, appear to be more inclined to share their information [63]. Participants have expressed that having print and web-based versions of programs would help mitigate these concerns [50]. However, it has also been reported that both adolescent girls and boys are more comfortable with filling out web based surveys of alcohol and tobacco use, when compared to filling out printed surveys, reporting higher and possibly more accurate rates of use [63]. For example, girls have reported a younger age of drinking onset when filing out a web-based survey when compared to a printed version. This may reflect the differing concerns of confidentiality and privacy as they relate to age, gender, and access to technology.

### 5.5. Furthering the Efficacy of Gender and Technology-Based Interventions

A re-occurring challenge of many substance use prevention and health promotion programs was the challenge of not only shifting attitudes, but behaviours as well. Some studies have reported behavioural changes. For example, studies have shown that education-based programs, which are central to a health promotion approach, have resulted in significant changes in girls’ behaviours towards substance use [47,48]. However, for boys and men, reported behavioural change was often less common than it was for girls and women [33,35,43]. A central issue to changing behaviours has been connected to the need for more longer term programs—an issue that some technology-based programs may find solutions to through the ability to provide wide-reaching interventions on relatively low cost platforms [43].

## 6. Limitations

The purpose of this review was to provide an examination of how technology-based substance use prevention approaches have integrated gender-informed health promotion and harm reduction, and how sex, gender and intersecting factors of age, race, and socioeconomic status related to substance use are, or are not, being achieved in these efforts. To identify the breadth of the literature on this topic, provide descriptive detail of the types of interventions that have been conducted, and to identify patterns and prevalence of use, a narrative review was conducted. Only four unique gender-transformative interventions were examined in the quantitative studies available, all of which included samples of only girls and/or women. Quantitative studies also varied in inclusion or measures of effect sizes, and thus there was an overall lack of data availability and consistency to conduct a meta-analysis that examined gender-comparative effects of gender-informed interventions. Future research in this field would benefit from more quantitative research that includes gender comparators, including trans and gender diverse individuals. This would also facilitate meta-analyses of the effects of gender-transformative interventions.

## 7. Conclusions

There has been a substantial amount of research conducted on the effects that sex and gender factors have on substance use and technological approaches to substance use prevention and health promotion. Compared to gender—and culturally-blind programming, the studies gathered and examined, demonstrated that gender—and culturally-tailored programming is more desired and effective at raising awareness and demonstrating measurable attitudinal and behavioural change related to substance use. And yet, few programs have incorporated sex and gender evidence into their program design, delivery and specific audiences to be reached, and fewer have analyzed the effectiveness of programs developed specifically through a SGBA+ lens. It is evident that there is still a considerable way to go to link the two and further incorporate sex, gender, and gender-transformative approaches in technology-based substance use prevention and health promotion efforts. Future research on this topic would benefit from additional quantitative analyses that incorporate gender comparisons to further identify effective gender-informed and transformative approaches.

The research has shown that compared to men, women tend to prefer technology-based substance use interventions, show higher substance use rates, and find these interventions more helpful. Generally, both men and women prefer technology-based interventions versus non technology-based interventions. Further, gender-specific and culturally-specific substance use programming is a preferential method for boys and girls. Programs that are educational and provide opportunities for self-reflection using personalized normative feedback and/or critical thinking are particularly successful. This research review demonstrates that health promotion-based technological approaches have some of the largest effects on women and girls—particularly in studies that focused on girls and reducing alcohol use and related harms. There is significant opportunity to partner with young men and women, and with technological platform designers, to increase the use, reach and effectiveness of health promotion, substance use prevention and brief interventions that take gender into account and advance gender equity.

With the advancements of technology-based health promotion and prevention efforts, there is a great opportunity to modify technological approaches to respond to the evidenced gaps, echoing the movement and progress of global eHealth and mHealth initiatives. These interventions have the opportunity to respond to shifts in technological engagement, reaching across generations with the goal and ability of streamlining the efficiency of substance use prevention and harm reduction, while keeping in mind how gender-transformative interventions can lead to successful health outcomes.

## Figures and Tables

**Figure 1 ijerph-17-00992-f001:**
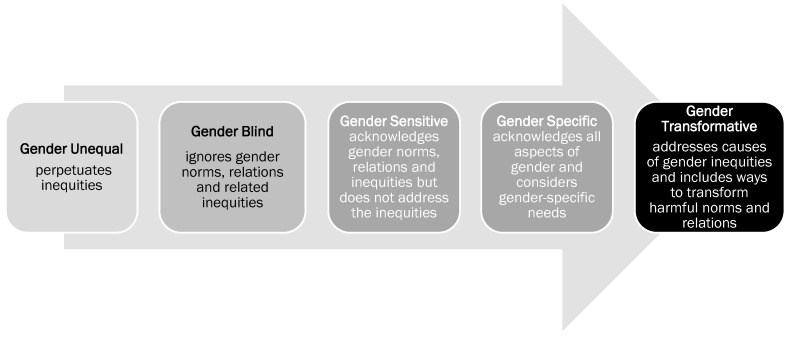
Gender Integration Continuum, reproduced adaptation with permission from Greaves, L., Pederson, A., Poole, N., Making it Better: Gender Transformative Health Promotion; published by Women’s Press, 2014 [14] (p. 22).

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
