# Peer review of "Technology-Based Substance Use Interventions: Opportunities for Gender-Transformative Health Promotion"

_ijerph, 2020, doi:10.3390/ijerph17030992_

Round 1
Reviewer 1 Report
This paper was a review of technology-based interventions for substance use with a focus on whether the intervention was blind or incorporated gender. The authors conclusions from their review were that technology-based interventions have some effectiveness particularly for women, and that such technology-based interventions for substance use may be helpful for populations where there is stigma about substance use. The authors have done impressive work reviewing a literature the spans across disciplines. There are a few areas, however, where the manuscript could be strengthened.
The main issue to me was that there seemed to be a bit of a disconnect between the opening of the paper and the main review. The authors make a compelling argument for why the continuum of gender-responsiveness of programs is important; however, in the review this seemed to not be emphasized as much. From the introduction, I thought that the authors would categorize studies along the continuum as a way to organize their review. This would require more work and perhaps double coding for reliability, but it may be a really helpful way for readers to organize this literature.
The second main issue is the lack of quantitative summaries of the reviewed articles. It’s hard to know how effective interventions were/are without measures of effect size. I would think the importance of this article would be greatly enhanced if it were turned into a meta-analysis. For instance, based on my point above, the authors could categorize interventions based on their gender-responsiveness and then calculate the effect sizes within each category (perhaps also by technology type). This could potentially be used to show that more gender-responsiveness interventions have bigger effects. It’s possible that there may not be enough effects in each category. Still, there’s an argument for at least having a table or graphic with effect sizes.
In general, I would like more information about the reviewed studies. I was particularly interested in the gender composition of the samples. It seems like this could be put into a table along with the effect sizes and some brief description of the intervention. Having this type of information available increases the value of this review.
The paper primarily seems to stick with the binary conceptualization of gender. I was curious where non-binary folks would fit into this. Is this a knowledge gap in the literature?
A minor thing that I noticed was the incorrect form of elicit was used a few times. The authors used illicit, which means illegal, when they mean elicit (e.g, on line 116 “to illicit change” should be “elicit change”).
Author Response
This paper was a review of technology-based interventions for substance use with a focus on whether the intervention was blind or incorporated gender. The authors conclusions from their review were that technology-based interventions have some effectiveness particularly for women, and that such technology-based interventions for substance use may be helpful for populations where there is stigma about substance use. The authors have done impressive work reviewing a literature the spans across disciplines. There are a few areas, however, where the manuscript could be strengthened.
Thank you very much for taking the time to review our manuscript.
The main issue to me was that there seemed to be a bit of a disconnect between the opening of the paper and the main review. The authors make a compelling argument for why the continuum of gender-responsiveness of programs is important; however, in the review this seemed to not be emphasized as much. From the introduction, I thought that the authors would categorize studies along the continuum as a way to organize their review. This would require more work and perhaps double coding for reliability, but it may be a really helpful way for readers to organize this literature.
Subsections of section 4 have been removed/integrated into section 5 to avoid disconnect. Additional connecting sentences have also been added to section 5 subsections.
Not all studies were examining an intervention, rather they gathered gender-based preferences in regards to technology and substance use, or simply provided gender disaggregated data. In addition, many studies only used single-gender groups and did not include a comparator. The authors therefore felt it would be more accurate to distinguish the papers based on technological interventions, and within that, identify how gender has been incorporated (or not) overall. This allowed for a broader lens of what literature addressed gender and/or sex at all, and the ability to comment on the varying ways these topics were addressed. The authors also felt it important to distinguish how differences in technology have differing effects in regards to access and inclusion based on multiple intersecting social factors (i.e. age, race, ethnicity in addition to gender). Additional introductory sentences have been added to the following lines for further clarification: lines 120-132.
The second main issue is the lack of quantitative summaries of the reviewed articles. It’s hard to know how effective interventions were/are without measures of effect size. I would think the importance of this article would be greatly enhanced if it were turned into a meta-analysis. For instance, based on my point above, the authors could categorize interventions based on their gender-responsiveness and then calculate the effect sizes within each category (perhaps also by technology type). This could potentially be used to show that more gender-responsiveness interventions have bigger effects. It’s possible that there may not be enough effects in each category. Still, there’s an argument for at least having a table or graphic with effect sizes.
Because the articles varied between quantitative, qualitative and descriptive approaches, and had different measures of program effectiveness (i.e. varying substances, measures of mental health, relational models), the authors chose a narrative review approach to provide an introductory global view of this topic, and to discuss varying measures of program effectiveness. Further, this approach allowed the researchers to focus on the gaps in the research. In addition, as mentioned in the previous comment, not all studies were examining an intervention specifically, rather they gathered gender-based preferences in regards to technology and substance use, and the authors extracted the relevant data. Many studies only used single-gender groups and did not include a comparator.
The following categories have been added to the existing table: study design, research aim, population (including gender compositions), intervention/procedure, key outcomes, and authors’ assessment of gender inclusion level based on gender continuum.
A sentence has been added to the following lines to add to future research considerations: Line 531-533.
In general, I would like more information about the reviewed studies. I was particularly interested in the gender composition of the samples. It seems like this could be put into a table along with the effect sizes and some brief description of the intervention. Having this type of information available increases the value of this review.
The following categories have been added to the existing table: study design, research aim, population (including gender compositions), intervention/procedure, key outcomes, and authors’ assessment of gender inclusion level based on gender continuum.
The paper primarily seems to stick with the binary conceptualization of gender. I was curious where non-binary folks would fit into this. Is this a knowledge gap in the literature?
This is a knowledge gap in the literature. The larger scoping review, from which the literature for this review was derived was inclusive of women, girls, men, boys, trans and gender diverse people of all ages and demographics. In the case of the technology-based substance use intervention, trans and gender diverse people were not represented in the literature.
This gap is addressed on lines 274-275. A paragraph has also been added to the following lines to further address this gap: lines 458-467.
A minor thing that I noticed was the incorrect form of elicit was used a few times. The authors used illicit, which means illegal, when they mean elicit (e.g, on line 116 “to illicit change” should be “elicit change”).
Changes have been made, and sections have subsequently been removed in section 4 reformatting.
Reviewer 2 Report
There is an interesting paper addressing the role and the need to include the gender differences in the use of technology-based interventions to reduce the use of substances.
There are several points.
Introduction:
When authors comment gender differences in substance use, it could be interesting to introduce how non-technology interventions address it. There are other examples that could be included, related to both drug and non-drug addictions. Introducing this point of view could enhance the applied relevance of the manuscript.
Methods:
Why authors have excluded psychostimulants of their research?
Results:
Before part 4, it needs to introduce Results. Why these results are not organized as the categories obtained: prevention, harm reduction and brief intervention. Only this last one was considered. I think that a brief paragraph explaining how results are presented could improve the manuscript reading.
Page. 6 line 272: GO-KO is stated instead of GO-KA.
In some parts there no exists any comment about gender differences in technology-based interventions. If in this case there are no obtained it should be stated. For example, in the 5.6. Phone Delivered Interventions. A comparison between male and female was not explained. In this case, only results regarding women are exposed.
I miss a global paragraph summarizing main gender differences obtained in the review.
Discussion & Conclusions
Page9, line 373. Check writing.
Page 9. Line 398. I do not understand the parenthesis after girls’ programs.
A more detailed applied use should be include to these parts, incorporating suggestions of how these types of interventions should be in order to contemplate gender differences.
References
It is possible to introduce some more new literature. It is a field with great level of research and very new papers are expected.
Author Response
There is an interesting paper addressing the role and the need to include the gender differences in the use of technology-based interventions to reduce the use of substances.
Thank you very much for taking the time to review our manuscript.
There are several points.
Introduction:
When authors comment gender differences in substance use, it could be interesting to introduce how non-technology interventions address it. There are other examples that could be included, related to both drug and non-drug addictions. Introducing this point of view could enhance the applied relevance of the manuscript.
How non-technology-based interventions address gender in substance use is addressed in the first paragraph (lines 29-39).
Methods:
Why authors have excluded psychostimulants of their research?
Information in this review is derived from an original scoping review research project on sex, gender and four substances (alcohol, cannabis, opioids and tobacco). The four substances were chosen for their current critical concern in Canada that have differential sex and gender impacts (this sentence has been added to lines 81-82). However, as noted in line 95, ‘multiple substances’ and ‘substance use generally’ have been included. Technology-based interventions that incorporate gender and/or sex primarily fell into these categories.
Results:
Before part 4, it needs to introduce Results. Why these results are not organized as the categories obtained: prevention, harm reduction and brief intervention. Only this last one was considered. I think that a brief paragraph explaining how results are presented could improve the manuscript reading.
Sections 4 and 5 encompass the results of the review, and what literature was available within the methods described. All categories of prevention, harm reduction, and brief intervention have been examined and considered. They are discussed throughout the results within the categories of technology-based interventions. ‘Brief web-based interventions’ was a common technology-based category used within the literature and therefore was used in this review. This type of intervention encompasses a harm-reduction approach. An introductory sentence has been added to lines 199-201 for clarification.
Additional introductory sentences have been added to the following lines for further clarification: Lines 120-132.
Page. 6 line 272: GO-KO is stated instead of GO-KA.
This has been changed (see line 330).
In some parts there no exists any comment about gender differences in technology-based interventions. If in this case there are no obtained it should be stated. For example, in the 5.6. Phone Delivered Interventions. A comparison between male and female was not explained. In this case, only results regarding women are exposed.
Not all interventions contained a sample of individuals of multiple genders, as was the case with gender-informed phone delivered interventions. Due to the limited research on this topic, this is why a narrative review was used, to explore the literature broadly and identify gaps. Sentences have been added to the following lines to further clarify: lines 410-416.
The following categories have also been added to the existing table: study design, research aim, population (including gender compositions), intervention/procedure, key outcomes, and authors’ assessment of gender inclusion level based on gender continuum.
I miss a global paragraph summarizing main gender differences obtained in the review.
A global paragraph of the main gender differences obtained in the review has been described on lines 521-533.
Discussion & Conclusions
Page9, line 373. Check writing.
Unclear on the issue.
Page 9. Line 398. I do not understand the parenthesis after girls’ programs.
This was to identify the connection between education-based approaches and health promotion, which is a central focus of the paper. I have reformatted to use commas in hopes to add clarity (see line 514).
A more detailed applied use should be include to these parts, incorporating suggestions of how these types of interventions should be in order to contemplate gender differences.
Suggestions are included in the closing paragraph from lines 534-545 i.e. using health-promotion based programs, particularly education-based programs for girls, critical thinking-based programs, as well as mHealth apps.
Additional sentences have been added to the following lines for further clarification: lines 537-539.
Additions to lines 531-533 have also been added to suggest future research to further examine gender comparisons to effectively comment on evidence-based application recommendations.
References
It is possible to introduce some more new literature. It is a field with great level of research and very new papers are expected.
This review article was derived from an original scoping review that contained literature inclusive of all harm reduction, health promotion/prevention and treatment interventions and programs available that include sex, gender and gender transformative elements in relation to opioids, alcohol, tobacco, cannabis, and multiple substances/substance use generally. Studies conducted with pregnant individuals, or that were pregnancy-focused were excluded from this original review as they were considered a unique population that garnered separate analysis (which the research team has conducted often on other research projects). Though there may be more recent research examining technology-based substance use interventions, or substance use interventions generally that incorporated sex and/or gender, the literature that combined both, and presented sex and gender-based results was limited. This is partially why the authors chose a narrative review approach, to provide a global review of the literature that has touched on this topic, and to identify gaps and directions for future research.
Round 2
Reviewer 1 Report
The authors have taken care to address several of my comments on their prior draft. In particular, they have added critical information to their table, reorganized sections, and pointed out the lack of representation of gender minorities.
I still think that this paper would have maximum impact if it incorporated some elements of meta-analysis. The authors make a somewhat compelling argument as to why they disagree that the informational value would be increased. Perhaps a middle group is to add the individual study effect sizes to the Table for quantitative studies. This would allow researchers and practitioners to consult the table and get a sense of how these interventions compare against each other. It would also facilitate future meta-analytic work when there are more studies in each domain.
Author Response
The authors have taken care to address several of my comments on their prior draft. In particular, they have added critical information to their table, reorganized sections, and pointed out the lack of representation of gender minorities.
Thank you very much for taking the time to review the manuscript and provide critical feedback for improvement.
I still think that this paper would have maximum impact if it incorporated some elements of meta-analysis. The authors make a somewhat compelling argument as to why they disagree that the informational value would be increased. Perhaps a middle group is to add the individual study effect sizes to the Table for quantitative studies. This would allow researchers and practitioners to consult the table and get a sense of how these interventions compare against each other. It would also facilitate future meta-analytic work when there are more studies in each domain.
Heeding your advice, we have reviewed the potential for adding the effect sizes of the quantitative papers to Table 1. This paper was conceptualized as a narrative review and the aim was to critically analyze the use of gender-informed technology-based approaches in substance use prevention and health promotion interventions. Therefore, the assessment of the efficacy or effectiveness of the outcomes reported in the included papers is beyond the scope of this paper. Moreover, within the quantitative papers, there were different outcomes that the authors measured (i.e. varying substance use patterns and prevalence of use, reductions of use for different substances, consequences of a variety of substances used, mental health indicators, readiness to change, communication, attitudes, self efficacy, refusal skills, among others). In addition, not all have measured effect size, and there is inconsistency with the effects being measured. Study groups and type of interventions were not homogenous and comparable. Not all quantitative papers have included gender-transformative interventions, or have measured the effects of a program based on gender/included a gender comparator. This manuscript is particularly interested in the outcomes for gender and technology, and gender-transformative interventions.
We believe qualitative, descriptive and mixed-methods approaches add significant value to this examination, and use of only the few quantitative articles that provide an effect size in relation to gender, substance use, and technology-based interventions would remove the possibility for ample content and detail in the review. Particularly as we are interested in gender-transformative approaches, there are only four unique gender-transformative interventions examined in the quantitative studies, all of which have samples of only girls and/or women, which does not allow for a gender comparator effect measure. A summary of the articles that examined gender-transformative interventions is also outlined on lines 121-125.
To address your concern, a limitations section has been added to lines 410-424 to identify the restrictions we have laid out in our response, and the importance of using elements of this initial review for the potential to conduct a meta-analysis in the future.